# Underlying Mechanisms of Reductive Amination on Pd-Catalysts: The Unique Role of Hydroxyl Group in Generating Sterically Hindered Amine

**DOI:** 10.3390/ijms23147621

**Published:** 2022-07-10

**Authors:** Zeng Hong, Xin Ge, Shaodong Zhou

**Affiliations:** 1Zhejiang Provincial Key Laboratory of Advanced Chemical Engineering Manufacture Technology, College of Chemical and Biological Engineering, Zhejiang University, Hangzhou 310027, China; 12128008@zju.edu.cn; 2Institute of Zhejiang University—Quzhou, 78 Jiuhua Boulevard North, Quzhou 324000, China; 3School of Chemical and Material Engineering, Jiangnan University, Lihu Avenue 1800, Wuxi 214122, China

**Keywords:** sterically hindered amine, reductive amination, competing mechanisms, in situ ATR–FTIR analysis, Pd-catalyst

## Abstract

Pd nanospecies supported on porous g-C_3_N_4_ nanosheets were prepared for efficient reductive amination reactions. The structures of the catalysts were characterized via FTIR, XRD, XPS, SEM, TEM, and TG analysis, and the mechanisms were investigated using in situ ATR–FTIR spectroscopic analysis complemented by theoretical calculation. It transpired that the valence state of the Pd is not the dominating factor; rather, the hydroxyl group of the Pd(OH)_2_ cluster is crucial. Thus, by passing protons between different molecules, the hydroxyl group facilitates both the generation of the imine intermediate and the reduction of the C=N unit. As a result, the sterically hindered amines can be obtained at high selectivity (>90%) at room temperature.

## 1. Introduction

Amines constitute an indispensable class of chemicals that are widely used as the raw materials or intermediates in the laboratory and in industry to prepare value-added chemicals, such as pharmaceuticals, agrochemicals, and biomolecules [1,2]. To date, numerous organic methodologies for amine production have been reported, such as the aminolysis of haloalkanes [3], the reaction of N-chloro dialkylamines with alkyl Grignard reagents [4], Buchwald–Hartwig and Ullman-type C–N cross-coupling reactions [5,6], as well as reductive amination. The amination of alcohols via so-called borrowing hydrogen (BH) is another important method which is mediated by rare noble-metal catalysts, based on Ru or Ir mostly [7]. Among all these methods, catalytic reductive aminations using molecular hydrogen as the reductant continue to be in the spotlight of both academic and industrial interests, due to their high atom economy and low pollution [8]. As to the catalysts required, in addition to precious metals such as Pd, Pt, Ru, and Rh [9], continuous efforts are being made on employing earth-abundant metals, such as Ni [10], Co [11], and Fe [12] as the active center. By this means, various amines, such as primary, secondary, and tertiary amines with less steric hindrance, can be prepared under moderate conditions [9].

In spite of the numerous studies reported previously on reductive aminations, the associated mechanisms are still disputable and remain ambiguous. It is generally believed that the transformation starts from the formation of imine upon condensation of the carbonyl group with ammonia or amine, followed by the reduction of imine [8]. However, a previous study pointed out that the imine intermediates do not form in the reductive amination with secondary amines [13]. For tertiary amines, the amine forms either by direct hydrogenolysis of the hemiaminal [13,14] or dehydration to enamine followed by hydrogenation [15]. Most likely, the preference of the reaction pathway depends on both the structure of the reactants and the reaction conditions.

In addition to the ambiguous mechanisms, reductive amination is yet limited for the synthesis of sterically hindered tertiary amines such as non-nucleophilic base and 1,2,2,6,6-Pentamethylpiperidine. In fact, as will be shown later in the Discussion section, we tried a series of commercially available catalysts that are proven to be excellent in mediating the production of most amines via reductive amination; however, most of these catalysts exhibit poor performance in the preparation of N,N-diisopropylbutylamine. Notably, Denis Chusov reported an atom-economical method for the synthesis of sterically hindered tertiary amines based on complementary Rh- and Ru-catalyzed direct reductive amination using carbon monoxide as a deoxygenating agent [16]. Indeed, sterically hindered amine is irreplaceable in organic synthesis and catalysis specifically as non-nucleophilic base [17], light stabilizers (HALS) [18], ligands [19], components of frustrated Lewis pairs [20], etc. As far as we know, the sterically hindered amines have been synthesized at industrial scale by reductive amination of carbonyl compounds with poor atom economy reagents such as borohydride reagents. Further efforts are still highly demanding for developing a more efficient preparation of sterically hindered amine via reductive amination with hydrogen.

On the other hand, Pd nanoparticles (e.g., Pd/C) have been proven to be efficient for reductive aminations with hydrogen, especially for the synthesis of tertiary amines [21]. Despite the fact that the classic Pd/C-H_2_ system is relatively less efficient in mediating the production of sterically hindered amine such as 1-cyclohexylpiperidine [22,23], Pd is still highly promising to complete this task benefiting from relatively strong Pd-C, Pd-H, and Pd-N interactions [24]. To tune the activity of the Pd catalysts, the influences of size effect and the support were investigated. The size distribution of the Pd/C catalyst can be controlled by quite a few strategies by tuning the reducing agents [25] and conditions [26], the concentrations of the stabilizing agent, Pd salt and precipitant [27,28,29], or introducing a second metal [30]. Support modification is another way to promote the catalytic activity of anchored metal nanoclusters [31]. The graphite-like carbon nitride (g-C_3_N_4_), characterized as an incompletely condensed, N-bridged “poly(tri-s-triazine)” polymer with lamellar structure, possessed unique physicochemical properties due to its appealing electronic band structure. The nitrogen functionalities on the surface might act as strong Lewis base sites and the π-bonded planar-layered configurations are expected to anchor the substrate and metal active species, making it a privileged candidate for hydrogenation reactions [32,33,34]. Moreover, treatments with various oxidants to introduce acidic sites on the surface of activated carbon may significantly improve the activity and selectivity of catalysts [35,36,37,38,39]. Up to now, however, it remains to be explored how the support affects selectivity for the production of sterically hindered amines via reductive amination.

In this article, we report an efficient Pd(OH)_2_/g-C_3_N_4_ catalyst for reductive amination. As compared to the previously reported Pd catalysts, the Pd-based nanoparticles prepared in this work are of higher activity and selectivity. By using a combination of structural characterization, in situ spectroscopic investigation, and theoretical calculation, the elaborate structure of the active center, as well as the root cause for the excellent performance of the Pd(OH)_2_/g-C_3_N_4_ catalyst, are revealed.

## 2. Results and Discussion

### 2.1. Catalytic Tests of the Prepared Palladium Catalysts

To evaluate the performance of the catalysts studied, the reaction of diisopropylamine with butyraldehyde under hydrogen atmosphere was selected as a model. Various catalysts, including commercially available ones and home-made ones, were employed, and the detailed results are listed in Table 1.

To start with, we took commercially 5.0 wt% Pd/ACs as a catalyst to investigate the effect of various conditions for the reaction. We found that the reaction can be proceeded at room temperature, and lower the ratio of amine to aldehyde (<2) or higher hydrogen pressure (>1.5 MPa) improves the selectivity of C=O reduction. Moreover, further prolongation of the reaction time (>4 h) will not bring about the increase in yield. Last but not least, we used methanol as the solvent with additive amount attributing to its high hydrogenation activity for reduction amination [40]. Thus, the optimal conditions for catalyst screening were determined. As shown in Table 1, classic inexpensive hydrogenation catalysts such as Raney Ni/Co/Cu failed to afford the desired product (Table 1, **Entry 1–3**). Next, for the noble-metal catalysts, only Pt/C and Pd/C successfully brought about the generation of diisopropylbutylamine, and Pd/C performed better (Table 1, **Entry 4–8**); however, the competing reduction of C=O was still unavoidable. In addition, atomically dispersed Pd did not give better performance (Table 1, **Entry 9**). Notably, when commercial 10.0 wt% Pd(OH)_2_/ACs was employed, the target product was obtained with 92% yield (Table 1, **Entry 10**). This result is consistent with a previous study on Pd(OH)_2_/ACs-catalyzed synthesis of tertiary amines [41]. However, when reducing the content of Pd loaded on ACs, the activity of the catalyst decreased dramatically: with ~1.0 wt% Pd, although preliminary nitric acid treatment and hydrothermal treatment of the ACs supports kept the Pd(OH)_2_/ACs catalyst at relatively good selectivity, the reaction rate dropped considerably (Table 1, **Entry 10**, **11c,d**). To further improve the performance of the Pd catalysts, we changed the support with g-C_3_N_4_, inspired by the finding of Wang et al. that phenol can be selectively reduced to cyclohexanone at Pd@g-C_3_N_4_ due to phenol being able to interact with the surface through the hydroxy group to form strong O–H···N or O–H···π interactions [34]. We considered that Pd(OH)_2_ tends to adsorb on basic sites of g-C_3_N_4_ by similar O–H···N or O–H···π interactions. Surprisingly, both the selectivity and the reaction rate maintain simultaneously on 1.1 wt% Pd(OH)_2_/g-C_3_N_4_. By contrast, however, when depositing reduced Pd nanoparticles on g-C_3_N_4_, the so-obtained Pd/g-C_3_N_4_ exhibits low catalytic activity (Table 1, **Entry 13**). Here, the valence state of Pd seems to be important. Thus, further examination on the performance of PdO species was carried out. In contrast to the Pd/support or the Pd(OH)_2_/support catalysts, much lower selectivity and yield are given by either PdO/ACs or PdO/g-C_3_N_4_ (Table 1, **Entry 15**, **16**). Therefore, it is not the high-valance state of Pd that matters; rather, the existence of the hydroxyl group in the Pd-oxide cluster substantially influences the reductive amination processes. Similarly, it has been found previously that the adsorption of hydrogen donors and acceptors on metal catalysts is species-dependent, thus offering opportunities for selectivity control in hydrogen transfer processes [42,43,44]. Finally, a satisfactory yield was obtained when using 4Å molecular sieve as dehydrating agent (Table 1, **Entry 12f**). Obviously, here, the reactivity of the system is sufficient, while the conversion is dominated by the reaction equilibrium.

### 2.2. Characterization of the Prepared Pd(OH)_2_/g-C_3_N_4_ and Other Palladium Catalysts

To reveal the root cause for the excellent performance of Pd(OH)_2_/g-C_3_N_4_, structural characterization is prerequisite. The Fourier-transform infrared (FT-IR) was performed. As shown in Figure 1, the absorption peaks at 1245, 1320, and 1408 cm^−1^ correspond to the aromatic C–N stretching vibrations, while 1567 and 1640 cm^−1^ are ascribed to the C=N vibrations. The peak at 808 cm^−1^ is assigned to the breathing mode of the triazine units (Figure 1c) [45,46]. The broad peak at 3000–3400 cm^−1^ is ascribed to the stretching vibration of N–H. When loading Pd to the g-C_3_N_4_, the intensity of these peaks does not change obviously but the intensity of the broad peak at 3000–3400 cm^−1^ increases (Figure 1a), which can be attributed to the O–H stretching of the Pd(OH)_2_. However, intensity of this broad peak decreases with treatment of hydrazine hydrate, indicating that the divalent Pd was reduced partially (Figure 1b). Accordingly, no obvious absorption peak is observed when Pd is loaded on the activated carbons treated by nitric acid or water (Figure 1d–f).

The phase structure and composition of the selected samples were investigated by X-ray diffraction (XRD). The obtained results are shown in Figure 2. The pattern of g-C_3_N_4_ is identified by peaks at 2θ = 27.8° and 12.8°, corresponding to the (002) crystallographic plane and (100) planes of in-planar tris-s-triazine structural packing motifs (Figure 2a,b) [47]. The broad peak at 25.1° indicated the typical amorphous structure of activated carbons (Figure 2c–f). The crystal structure of Pd is identified by the diffraction peaks at 2θ = 40.4°, 47.0°, and 68.6° (JCPDS 87-0645). These peaks can be assigned to the (111), (200), and (220) planes of Pd metal with a face-centered cubic (fcc) structure, respectively. However, no Pd or PdO characteristic peaks were detected for Pd(OH)_2_/g-C_3_N_4_, suggesting that Pd species may exist in the form of subcrystal or nanoparticles. A weak peak at 40.4° can be observed after reduction (Figure 2b). According to Scherrer’s formula [48] and the half-width of the Pd (111) peak, the calculated size of Pd NPs in Pd/g-C_3_N_4_ is 1.3 nm. When Pd is loaded on ACs, the location of the Pd (111) peak (at 40.4°) remains with a calculated size of Pd 2.4 nm, 3.7 nm, and 2.0 nm. It can be seen that the divalent Pd can be reduced by the reducing groups anchored on the surface of activated carbons during the impregnation process (Figure 2c–e) [49]. Additionally, both PdO/ACs and the commercial 10.0 wt% Pd(OH)_2_/ACs catalyst exhibited diffraction peaks at two-theta of 33.5°, 33.8°, 60.2°, and 71.4° in their XRD patterns (Appendix A), which can be indexed as the (101), (112), (103), and (211) diffractions of Palladium oxide (JCPDS 06-0515). Furthermore, the PdH_x_ phase was not observed in any of the XRD patterns, which is considered as the active species for hydrogenation reactions [50].

Next, XPS measurements were performed to explore the chemical properties of the surface and the electronic configurations of the active centers. The survey XPS spectrum in Figure 3a shows that C, O, N, and Pd elements coexist in 1.1 wt% Pd(OH)_2_/g-C_3_N_4_ sample. Figure 3b–d show XPS core level spectra of C 1s, N 1s, and Pd 3d of the 1.1 wt% Pd(OH)_2_/g-C_3_N_4_ sample. All lines in the XPS spectra are corrected with carbon C 1s at 284.8 eV. The peak at 284.8 eV in C 1s is attributed to surface adventitious carbon, whereas the peak at 288.2 eV corresponds to sp^2^-bonded carbon in N–C=N of g-C_3_N_4_ (Figure 3b). For N 1s, four peaks can be distinguished (Figure 3c). The signal at the binding energy of 398.5 eV is assigned to the pyridinic N (N–C=N), and the peak at 399.9 eV is assigned to the graphitic N. The peaks at 401.0 and 404.6 eV are attributed to the amino group (C–N–H) and some N–O species [33]. The spectra of Pd 3d (Figure 3d) present two doublet peaks, corresponding to the spin–orbital splitting of Pd 3d_5/2_ and Pd 3d_3/2_ for two types of Pd species. The peaks at 335–336 eV can be assigned to reduced Pd and the ones at 336–337 eV to palladium oxide or palladium hydroxides. It has been pointed out previously that Pd(OH)_2_ on carbon materials is a core–shell structure of C/PdO/OH/H_2_O [51]. The ratio of Pd^0^ to Pd^2+^ was about 31:69 using Gaussian/Lorentzian line shape approximations, suggesting that the Pd atoms in Pd(OH)_2_ are partially reduced during the preparation (see Appendix A). After treatment with hydrazine hydrate, the ratio of Pd^0^ to Pd^2+^ increases to 70:30 (Appendix A). Thus, the bivalent palladium species was considered as the main active component in the reaction. To our surprise, the content of pyridinic N decreased from 76% to 74% and the content of graphitic N increased from 16% to 19% when the catalyst was treated by hydrazine hydrate (see Figure 2c and Appendix A). Previous reports revealed that pyridinic coordination sites exhibited the highest metal-loading stability, whereas the graphitic-N coordination sites had the least stable one [33,52]. We infer that the Pd species may aggregate during reduction by hydrazine hydrate. On the other hand, N 1s signals of Pd/g-C_3_N_4_ shift 0.05 eV towards low binding energy, indicating that the interaction between the g-C_3_N_4_ and Pd is weakened after reduction. For Pd(OH)_2_/ACs, the one whose support is hydrothermally treated possesses more phenolic groups on the surface (Appendix A). Notably, the Pd 3d_5/2_ peak for the Pd species on the hydrothermally treated supports is 0.8 eV higher than that of the nitric acid treatment supports, as depicted in Appendix A, indicating the presence of a strong interaction between the support and Pd. According to the literature, Pd species can be stabilized by the deprotonation process between the Pd precursor and phenolic groups (−OH) to produce an oxygen anion (−O−) and facilitate the dispersion of Pd [39,53].

In order to identify the morphology of the Pd species on g-C_3_N_4_, scanning electron microscopy (SEM) and transmission electron microscopy (TEM) were performed for Pd(OH)_2_/g-C_3_N_4_. The SEM image (Figure 4a) demonstrates that the Pd(OH)_2_/g-C_3_N_4_ composite possesses a fluffy, wrinkled, and porous microstructure. Meanwhile, it can be seen from the TEM images (Figure 4c) that the Pd species were distributed throughout the g-C_3_N_4_ architecture uniformly without obvious aggregation, disclosing that the nanosheets are layered flakes and possess some defects and voids which provide abundant anchoring sites for Pd species. The average particle size of the Pd nanoparticles in Pd(OH)_2_/g-C_3_N_4_ is 2.11 nm, as determined by statistical evaluation of 100 particles in the TEM images. The Pd content of the Pd(OH)_2_/g-C_3_N_4_ and Pd/g-C_3_N_4_ sample determined by ICP-OES is calculated to be 1.1 wt% and 1.2 wt%.

Subsequently, according to the thermogravimetric (TG) analysis, for Pd(OH)_2_/g-C_3_N_4_, there is an initial weight loss ∼135 °C which is due to the removal of adsorbed water molecules within the system. After that, the Pd(OH)_2_/g-C_3_N_4_ sample is stable up to 500 °C and starts to decompose and decomposition is complete ∼680 °C with a residual weight of 2.0% (Figure 4b); this decomposing temperature is higher compared to Pd/g-C_3_N_4_ [54]. Alternatively, the thermal stability of the catalyst can also be improved by introducing hydroxyl groups to the support-activated carbons with hydrothermal treatment (Appendix A). Thus, the presence of the hydroxyl groups enhances the thermal stability of the catalyst. Furthermore, for the heterogeneous catalysis reaction, it is important to test the reusability of the catalyst. Thus, the reusability of the 1.1 wt% Pd(OH)_2_/g-C_3_N_4_ catalyst was investigated using the reductive amination of diisopropylamine with butyraldehyde as a model reaction. Moderate selectivity toward N,N-diisopropylbutylamine was obtained after four runs (Figure 4d), suggesting that this catalyst has receptable reusability.

### 2.3. Mechanism Studies

Upon addressing the structural features of different catalysts and recycling tests, the in situ ATR–FTIR spectroscopic analysis was employed to identify how the catalysts interact with the substrates during the reduction amination. The peaks of [*υ*(C=O)=1740–1780 cm^−1^] were firstly identified for all the spectra [55]. For commercial 1.0 wt% Pd/ACs, the imine species [*υ*(C=N)=1625 cm^−1^] with weak intensity was observed when acetaldehyde and ammonia were introduced to the in situ reaction cell [56]. This peak decreased but did not disappear when continuously introducing H_2_ (Figure 5a) into the system. Meanwhile, the [*υ*(C–N)] mode in the range of 1350–1280 cm^−1^ cannot be observed, probably due to its low intensity [57,58]. Note that there is no stretching vibration peak of water at 3200–3500 cm^−1^, indicating that the dehydration process may not require a catalyst [59]. It is also confirmed by the fact that ethanimine formed without a catalyst (See Appendix A). The C=N vibration was also observed when acetone and NH_3_ were introduced into the cell, and a peak around 3464 cm^−1^ was observed (Figure 5b). Considering the [*υ*(O-H)] stretching mode adjacent to the nitrogen atom; most likely, this hydroxyl group belongs to the hemiaminal intermediate [60]. The peaks at 1200 cm^−1^ and 1130 cm^−1^ were assigned to the deformation stretching of the C–N bond (Figure 5c) [57]. The imine mechanism thus prevails as well. By contrast, for the secondary amine diethylamine, a broad peak located at 3200–3600 cm^−1^ emerged only after H_2_ was introduced. This can be ascribed to the stretching vibration of adsorbed water, which was further proven by the peak in 1640 cm^−1^ (assigned to the H–O–H bending bands in product water) (Figure 5d) [59]. Meanwhile, the skeleton-stretching vibration of (-C_3_)N [*υ*((-C_3_)N)=1178 cm^−1^] is also identified [58], suggesting that the dehydration and hydrogenation processes occur simultaneously on the surface of the catalyst. As for the amination of diisopropylamine, the commercial 1.0 wt% Pd/ACs, Pd(OH)_2_/g-C_3_N_4_, Pd/g-C_3_N_4_, and PdO/g-C_3_N_4_ were compared (Figure 5e–h). For all catalysts, a weak peak, as discussed above, assigned to the stretching vibration of the hydroxyl group of the hemiaminal, appeared at 3487 cm^−1^, 3491 cm^−1^, 3474 cm^−1^, and 3483 cm^−1^, respectively, after introducing diisopropylamine and acetaldehyde to the in situ reaction cell. They were found as red-shift peaks due to the difference of surface binding or adsorption strength between the different Pd catalysts since the hydroxyl group vibration of the free hemiaminal is 3628 cm^−1^, according to DFT calculation at the B3LYP/6-31+(g) level. The peak of the water shows a completely different shape difference, which may imply a completely different catalytic performance on the home-made catalysts. Except for PdO/g-C_3_N_4_, all the other catalysts show the peaks of absorbed water, which is in line with the experimental results that home-made PdO/g-C_3_N_4_ is inactive. Moreover, no imine peak at the range 1620–1680 cm^−1^ was observed [56]. The identification of the product might be difficult because of the low intensity of the typical band at 1220–1270 cm^−1^ assigned for the skeleton-stretching vibration of (-C_3_)N [58].

Based on the in situ characterization, the reaction mechanisms were investigated by theoretical calculations. Considering the different performances of Pd/g-C_3_N_4_ and Pd(OH)_2_/g-C_3_N_4_, the valence state of Pd matters for the reductive amination of diisopropylamine. The generation of imines for different substrates was examined computationally, concerning both the hydrogen transfer and dehydration processes (see Appendix A). They are all kinetically less favorable due to high reaction barriers (>40 kcal/mol). However, when a second amine is introduced into the reaction complex, the subsequent reaction barrier for both hydrogen transfer and dehydration is lowered by 2 kcal/mol and 10 kcal/mol, respectively. Further enhancement in these two steps is achieved when a third amine is introduced (see Appendix A). It seems that dehydration from aldehydes and primary amines or ammonia is a multimolecule synergistic process. This result is in line with the in situ ATR–FTIR spectroscopic findings (see Appendix A). On the contrary, for imine hydrogenation, a catalyst is required since the reaction barrier is extremely high (>60 kcal/mol; see Appendix A).

Thus, the Pd(111) plane was used for modeling the Pd/g-C_3_N_4_ catalyst to understand the catalytic performance on Pd^0^ species. We first investigated the adsorption behavior of hydrogen on the Pd(111) surface, demonstrating that the hydrogen is preferentially absorbed on the bridge and hollow sites (see Appendix A). Furthermore, as shown in Figure 6, initially diisopropylamine and acetaldehyde are coadsorbed on the Pd(111) surface. The N–H bond activation of diisopropylamine takes place upon hydrogen transfer from the nitrogen atom to the oxygen atom via **TS1/2**, and C–N coupling takes place spontaneously (**IM1**→ → **IM2**). Species adsorption on the catalyst surface significantly lowers the reaction energy barrier, making it feasible under mild conditions (see Appendix A). Next, palladium hydride attacks the oxygen atom, thus breaking the C–O bond via **TS3/4**, which was identified as the rate-determining step (RDS). After that, **IM4** is generated, which further releases a water molecule to form **IM5**. Finally, by transferring a hydrogen atom from the Pd surface to N, an intact N,N-Diisopropylethylamine molecule is produced (**IM6**). This process is featured as the hydrogenolysis mechanism [61]. In addition to the pathways via hydrogenolysis, an alternative imine path was also investigated [62]. Specifically, after the imine intermediate is generated via **TS7/8**, hydrogen transfer from Pd to C affords the target product (**IM9** → → **IM****10**). This step is followed by further hydrogenation of the hydroxyl groups, thus generating a water molecule (**IM12**). Comparing the above two pathways, the latter is energetically more favorable. Furthermore, we examined theoretically the hydrogenation of the acetaldehyde on the Pd(111) surface. As shown in Appendix A, the H atoms are successively added to O and C in a step-wise manner via the sequence **IM13 → TS13/14 → IM14 → IM15 → TS15/16 → IM16**. Alternatively, the H atoms are successively added to C and O in a step-wise manner via the sequence **IM17 → TS17/18 → IM18 → IM19 → TS19/20 → IM****20.** Although the former is thermodynamically more favorable due to its high exothermicity (−81 kcal/mol), the latter is kinetically more preferred due to the low barrier for the hydrogenation process.

For Pd(OH)_2_/g-C_3_N_4_, considering the structure characterized previously [51], a Pd_8_O_12_H_8_ cluster was used to simulate the reductive amination processes. As shown in Figure 7, when diisopropylamine and acetaldehyde are coadsorbed on the clusters, the hemiaminal intermediate is formed spontaneously without a reaction barrier by simple hydrogen exchange between diisopropylamine and a hydroxyl group (**IM21** → **IM22**). On the contrary, the reaction barrier for forming a hemiaminal on the PdO cluster is extremely high (45 kcal/mol; see Appendix A), making PdO/g-C_3_N_4_ inactive. Here, the presence of the O–Pd–OH site is crucial. We first consider that the hydrogenolysis process occurs on the clusters which is similar to that on the Pd(111) surface. The transitional structure involving palladium hydride attacking the oxygen atom to form a water molecule was firstly located. However, the subsequent hydrogenation transition state structure could not be found (see Appendix A). After breaking the C–O bond (**IM23** → → **IM24**), the hydroxyl group on the catalyst surface serves as a “courier” to pass hydrogen. Thus, a water molecule forms first by hydrogen transfer from Pd to OH via **TS25/26.** Next, a hydrogen atom on the so-formed water molecule migrates to the imine carbon to form the product (**IM26** → → **IM27**). Here, we considered that the imine species can be polarized by a hydrogen bond formed between N and a hydrogen atom on the so-formed water, thus increasing the probability to generate a transition state for H addition. Finally, by transferring a second hydrogen atom from Pd OH, an intact water molecule forms again (**IM28**). This process is similar to phenol acting as a conduit to transfer a proton from the hydronium ion (with an accompanying charge transfer from the metal surface) to the basic carbonyl oxygen of the benzaldehyde via a PCET mechanism in the electrochemical hydrogenation (ECH) system, as reported by Udishnu Sanyal et al. [63]. Moreover, a thermodynamically more favorable pathway (−49 kcal/mol) was also found via the sequence **IM25** → **TS25/26** → **IM26** → **IM29** → **TS29/30** → **IM30**. Considering the difference in the reaction pathways presented in Pd(111) and the Pd cluster, the hydroxyl group is unique in that: (1) via barrierless (or quasi-) proton exchange, the formation of a hemiaminal is facilitated at the O–Pd–OH site; (2) the hydroxyl group serves as a hydrogen shuttle in the reduction of the imine unit concerted with the dihydrogen activation; (3) the Pd nanoclusters may be stably anchored on the basic sites of g-C_3_N_4_ by forming strong O–H···N or O–H···π interactions. In spite of the lower energy barrier for the hydrogenation of acetaldehyde (see Appendix A), the surface of the cluster preferentially adsorbs the hemiaminal intermediate by similarly forming O–H···N interactions which are generated spontaneously in advance. A high selectivity for reductive amination thus results, in line with the experimental findings. The probable mechanism for the generation of diisopropylethylamine is shown in Figure 1, and the Appendix A showed the intact outline for the generation of diisopropylethylamine catalyzed by a Pd_8_O_12_H_8_ cluster.

To examine the versatility of the reduction amination on Pd(OH)_2_/g-C_3_N_4_, various aldehydes and amines (mainly secondary ones) were used as the substrates (Table 2), especially for the ones with sterically hindered groups. For paraformaldehyde as a substrate, higher temperature (363 K) and more methanol (10 mL) were acquired to dissolve it. Other reaction conditions are similar with reported in Table 1. As for diisopropylamine and aliphatic aldehyde (**Entry 1,2**), the selectivity to the corresponding tertiary amine could reach 99 and 87%, respectively. Piperidines (**Entry 3–6**) gave a similar result to diisopropylamine (**Entry 1**). However, the selectivity of the target tertiary amine dropped as expected when acetaldehyde was employed due to its higher reactivity that can launch the reduction of the carbonyl group and aldol condensation (**Entry 7,8**) [22]. Apart from piperidines, the chain secondary amine (**Entry 9**) could also afford relatively high conversion and selectivity. Unfortunately, when aromatic amines such as diphenylamine and N-ethylaniline were employed (**Entry 10,11**), the selectivity and yield were significantly decreased due to the competing C–N activation processes [64]. On the contrary, dicyclohexylamine (**Entry 12**) could be transformed with 99% conversion and 99% selectivity. As for cyclohexylamine (**Entry 13**), the conversion and selectivity could reach 70% and 75%, respectively. The ^1^H and ^13^C NMR spectra of all products can be found in the Appendix A). 

## 3. Materials and Methods

### 3.1. Chemicals and Materials

Palladium dichloride (PdCl_2_), palladium oxide (PdO), and 10.0 wt% Pd(OH)_2_/ACs were purchased from Alfa Aesar. Diisopropylamine, butyraldehyde, activated carbon, melamine, cyanuric acid, methanol, and molecular sieve were obtained from Sinopharm Chemical Reagent Co., Ltd. Water was purified by ion exchange and used as deionized water. Other chemicals were of analytical purity and were used as received.

### 3.2. Characterization Techniques

The X-ray diffraction (XRD) patterns of the samples were recorded using a Shimadzu X-ray diffractometer (MAXima XRD-7000, the Japan) with Cu Kα radiation at 40 kV and 40 mA. The X-ray photoelectron spectroscopy (XPS) measurements were performed on a Thermo Scientific K-Alpha+ system using Al Kα radiation (1486.6 eV) under a base pressure of 2 × 10^−7^ Torr. A FEI QUANTA FEG 650 field emission scanning electron microscope (SEM) operated at 30 kV and transmission electron microscopy (TEM) using a HT-7700 (Tokyo, Japan) at 100 kV were used for determining the morphology of the as-synthesized Pd catalysts. The Pd loading in the materials was analyzed by an Agilent 720ES type inductively coupled plasma optical emission spectroscopy (ICP-OES) instrument. The FTIR spectra were recorded using a Thermo Scientific Nicolet iS50 FTIR spectrometer, and in situ FT-IR (ATR-FT-IR) spectra were recorded using the same spectrometer with ATR appendix. The thermal gravimetric analysis (TGA) was performed by using the Pyris 1 TGA thermogravimetric analyzer with a heating rate of 5 °C/min under nitrogen atmosphere. All products were determined by 7820A-5977B GC-MS (Agilent Technologies, CA, USA) using HP-5MS column. ^1^H NMR and ^13^C NMR were recorded on a Bruker AVANCE III 400 MHz (101 MHz for ^13^C), using deuterated chloroform (CDCl_3_) and tetramethylsilane (TMS, δ = 0) as internal reference. Chemical shifts are reported in parts per million (ppm) downfield and quoted to the nearest 0.01 ppm relative to the residual protons in the NMR solvent ((^1^H NMR: δ 7.26 ppm, ^13^C NMR: δ 77.16 ppm)), and coupling constants (J) are quoted in Hertz. GC analyses were performed on an Agilent 5973−6890 series gas chromatograph system (Agilent Technologies, CA, USA) equipped with a flame ionization detector (FID). All the separations were performed on a weakly polar capillary column SE-30.

### 3.3. Preparation of the Supports and Palladium Catalysts

#### 3.3.1. Preparation of g-C_3_N_4_ Support

Bulk g-C_3_N_4_ was synthesized by a thermal treatment of a mixture of melamine and cyanuric acid with a weight ratio of 1:1 [65]. All chemicals used in the preparation were of analytical grade without further treatment. Typically, 10 g above precursor powder was placed in an alumina crucible without a cover. Then, the crucible was placed in the middle section of a tube. After vacuuming the tube, argon gas was continuously fed in with a flow rate of 50 mL min^−1^ during the thermal treatment. The sample was heated to 120 °C at a rate of 10 °C·min^−1^, and maintained at this temperature for 20 min. The mixture was then heated to 550 °C and calcinated for 5 h. After cooling to room temperature, ~1700 mg of bulk g-C_3_N_4_ was obtained.

#### 3.3.2. Pretreatment of Activated Carbon

The pretreatment of ACs has been reported elsewhere previously [38,39]. The activated carbon was pretreated with 0.4N of HNO_3_ at 353 K for 4 h, was then washed with distilled water until pH = 7 after cooling to room temperature, and eventually dried in a vacuum at 323 K overnight. For hydrothermal treatment, 40 mL of deionized water was introduced into a 100 mL high-pressure autoclave; then, it was heated to 453 K and the pressure was self-generated, held at that temperature for 2 h, and subsequently cooled to room temperature. The treated ACs were filtered from the slurry and dried at 323 K overnight.

#### 3.3.3. Preparation of the Pd/ACs Catalyst

Pd/ACs catalysts were synthesized by wetness impregnation method. First, 30 mg PdCl_2_ was dissolved into 10 mL water with HCl mixture solution (volume/volume = 10:1). Then, the treated ACs (the ratio of ACs to water was 1 g/10 mL) were introduced into the PdCl_2_ solution and the slurry was vigorously stirred at 353 K for 4 h; then, the slurry solution pH of 8−9 was reached by dropwise addition of KOH aqueous solution (10%). Eventually, the precipitated Pd(OH)_2_ supported on ACs was reduced by 50% hydrazine hydrate at room temperature, filtered, and dried in vacuum at 323 K overnight.

#### 3.3.4. Preparation of the Pd(OH)_2_/g-C_3_N_4_ and Pd(OH)_2_/ACs Catalysts

The Pd(OH)_2_/C_3_N_4_ and Pd(OH)_2_/ACs catalysts were synthesized with similar method without further reduction. First, 30 mg PdCl_2_ was dissolved into 10 mL water with HCl mixture solution (volume/volume = 10:1). The PdCl_2_ solution pH of 8−9 was reached by dropwise addition of KOH aqueous solution (10%). Then, the slurry was mixed with 1 g g-C_3_N_4_ and vigorously stirred at 323 K in a three-neck flask for 4 h, then filtered and dried in vacuum at 323 K overnight. This was the same procedure as Pd(OH)_2_/ACs catalyst. PdO/ACs was prepared using PdO as precursor instead of the Pd(OH)_2_ deposition.

### 3.4. General Procedure for the Preparation of Sterically Hindered Amine and Recycling Experiments

Diisopropylamine (0.1 mol) and butyraldehyde (0.05 mol), 5.0 wt% catalyst (0.2 g), and methanol (2.5 mL) were mixed into 100 mL high-pressure autoclave at 30 °C. Then, 1.5 MPa H_2_ was fed into the reaction mixture. The mixture was stirred at 1000 rpm at 30 °C for 4 h. After the reaction, the rest of H_2_ was discharged. The conversion of the butyraldehyde and the sterically hindered amine yields were determined by GC with triethylamine as the internal standard. After the reaction, the catalyst was filtered and exhaustively washed with methanol, and dried in vacuum at 323 K for 12 h. The collected catalyst was used for the next run under the same conditions. Other cycles were repeated following a similar procedure. All the products were separated and purified by vacuum distillation or silica gel column chromatography. For **Entry 12** in Table 1 and **Entry 1–3, 5, 9, 12** in Table 2, the products were separated via vacuum distillation in a rotary evaporator; for **Entry**
**7,8, 13** in Table 2, the products were separated via vacuum distillation in a rotary evaporator to remove the solvent and then vacuum rectification with a home-made rectification column using spring packing; for **Entry 4, 6, 10, and 11** in Table 2, the products were purified by silica gel column chromatography.

The efficiency of the catalyst was characterized by calculating the conversion (*Conv*.%) of butyraldehyde (*BA*) based on the following equation (Equation (1)):(1)$$Conv.\%=\frac{n{\;}_{consumed}\;BA}{n{\;}_{initial}\;BA}\cdot 100$$

Sterically hindered amine (*AN*) yield (*Y*%) was also calculated as follows (Equation (2)):(2)$$Yield\%=\frac{n{\;}_{formed}\;AN}{n_{\;theoretical}\;AN}\cdot 100$$

Furthermore, *AN* selectivity (*Sel*.%) was calculated according to the following equation (Equation (3)):(3)$$Sel.\%=\frac{n{\;}_{formed}\;AN}{\sum n{\;}_{products}}\cdot 100$$

### 3.5. General Operation Procedure for the In Situ FT-IR Spectra

In situ FT-IR spectra to investigate the adsorption of different acetaldehyde and diisopropylamine on Pd catalyst were recorded on a Nicolet iS50 spectrometer equipped with a cell fitted with BaF_2_ windows and an MCT-A detector cooled with liquid nitrogen. The spectrum was collected at a resolution of 4 cm^−1^ with an accumulation of 32 scans in the range of 4000–700 cm^−1^. Here, the interaction of acetaldehyde and diisopropylamine was taken as an example (Figure 5f), and the catalyst sample (25 mg) was filled in priorly into a self-supported wafer. Then, the sample wafer was pretreated under a flow of N_2_ for 30 min at room temperature to remove the impurities absorbed on the surface. The background was collected at room temperature under a flow of N_2_. Then, liquid diisopropylamine (2 mL) filled into a glass tube was firstly vaporized by heating belt flowing by N_2_ to sample wafer. Considering that pursuing a spectrum with acceptable adsorption intensity and liquefaction of substrates on the surface of the catalyst should be avoided, the temperature of the sample wafer was chosen as 323 K for all the catalysts. The spectra of adsorption of diisopropylamine were recorded per minute until stable spectra were obtained. Then, acetaldehyde (2 mL) was vaporized by heating belt to sample wafer as well. The spectra of coadsorption of diisopropylamine and acetaldehyde were recorded per minute until stable spectra were obtained. Finally, the system was purged with a flow of hydrogen instead of N_2_ to obtain the spectra of aldehyde/amine/H_2_ mixture. After a stable spectrum was obtained, the system was purged with a flow of N_2_ again, and the desorption spectra were recorded toward the desorption time until there was no change in the band intensity. Other catalysts were then tested in a similar manner.

### 3.6. Computational Details

The structural optimization and the frequency analysis were performed at the GFN-xTB level of the xTB package [66,67]. Stationary points were optimized without symmetry constraint, and their nature was confirmed by vibrational frequency analysis. Unscaled vibrational frequencies were used to correct the relative energies for zero-point vibrational energy (ZPVE) contributions. Intrinsic reaction coordinate (IRC) [68,69,70] calculations were also performed to link transition structures with the respective intermediates; to achieve this, a Gaussian interface to the xTB code “gau_xtb” was employed [71].

## 4. Conclusions

In summary, a highly selective Pd catalyst designed specifically for the preparation of sterically hindered amine was reported. The selectivity of diisopropylbutylamine was up to 97%, and the yield of product amounted to 73%. The catalyst was characterized using FTIR, XRD, XPS, SEM, TEM, and TG, indicating that the active metal was well dispersed on g-C_3_N_4_ and stabilized by the hydroxyl group. The in situ ATR-FTIR measurement, together with the theoretical calculations, reveal that the generation of a key intermediate hemiaminal was facilitated by the proton exchange process, and the preferential adsorption of the hemiaminal enabled the selective reduction amination of diisopropylamine to sterically hindered amine. Here, the bifunctionality of the hydroxyl group mattered. An imine mechanism was thus justified for the generation of sterically hindered tertiary amines. Our work shed light on the catalytic mechanisms of reductive amination; in particular, the critical role of the hydroxyl group in the generation of sterically hindered tertiary amine was revealed. Hopefully, this may be instructive for the construction of amine moieties in specific molecules.

## Data Availability

Additional figures are available in the Appendix A.

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
