# Peer review of "Underlying Mechanisms of Reductive Amination on Pd-Catalysts: The Unique Role of Hydroxyl Group in Generating Sterically Hindered Amine"

_ijms, 2022, doi:10.3390/ijms23147621_

Round 1

Reviewer 1 Report

In the present manuscript titled as “Striking the Underlying Mechanisms of Reductive Amination on Pd-catalysts: The Unique Role of Hydroxyl Group in 3 Generating Sterically Hindered Amine” by Zeng Hong et al. showed the underlying reaction mechanism of reductive amination using Pd(OH)2 catalyst and highlight the unique feature of hydroxyl groups to enhance the activity and selectivity to desired product. Catalysts were characterized in detail with various microscopic and spectroscopic characterization. Underlying reaction mechanism corresponding to reductive amination was established by in-situ IR as well as theoretical calculation. 

While the current results showed in the manuscript is important, it needs significant improvement before it can be considered for the publication. I also suggest authors to improve the discussion in some of the section as mentioned below.  

Please find below the comments and suggestions that needs to be addressed in detail. 

1.     I suggest authors to remove the phrase “Striking the” from the title.  

2.     What precursor did they use for PdO synthesis? 

3.     Table 1, line 12 – why there are two different values for conversion, selectivity, and yield?

4.     How was the selectivity to product resulted upon reduction of butyraldehyde? It seems Pd exhibited lower selectivity to -C=O reduced product. Did authors have any explanation in that? Chemoselectivity of Pt and Pd towards C=O and C=C reduction is widely reported in literature and authors should consider that. 

5.     In Table 1, comparing line 12 and 13, it is apparent that when g-C3N4 support was used, activity of Pd(OH)2 was higher compared to Pd, however, the reverse is true when ACs was used as support. Can authors provide a rational in this regard? 

6.     Page 5, line 212 – I am not sure what authors meant for existence of H atom on Pd-oxide cluster? I think it should be the hydroxyl anion that are important for the activity. 

7.     Page 6, line 217 – I do not find any entry corresponding to 12f. authors should check that

8.     In Figure 1, authors mentioned that upon loading Pd, an increase in -OH stretching vibration was noted. In discussion, they should explicitly mention whether it is due to Pd or Pd(OH)2

9.     The presentation of Figure 1 is very confusing, there are both color code and a, b, c, d etc. which are very different. Authors should clarify how to follow Figure 1. 

10.  In figure 2c and 2d, what is the meaning of ACs-N and ACs-H? with AC as support, all the Pd(OH)2 catalyst show a peak corresponding to Pd(0) which suggested some reduction took place. Why is it so? Also, why the intensity is different across the catalysts?

11.  The smaller particles showed in figure 4c are corresponding to Pd or Pd(OH)2 clusters? Did authors know what underlying reason for catalyst deactivation is, i.e., lowering both conversion and selectivity

12.  Why the authors used acetaldehyde instead of butyraldehyde for in-situ IR experiments?

13.  Authors should move figure 7 to the supporting information

14.  Line 425, it should be -OH instead of O-

15.  In my opinion, hydrogen bonding is one of the major factor why Pd(OH)2 shows much higher activity compared to Pd. I suggest authors to consider the following paper and the back reference included in it. (Angew. Chem. Int. Ed. 10.1002/ange.202008178)

16.  In Table 2, what are the major side products obtained when the selectivity to desired product is low

17.  Line 439-440, What authors meant hydroxyl group serves as a hydrogen transition state? Also, they should explain how presence of hydroxyl group facilitates dihydrogen activation. 

Author Response

Dear Editor,

Many thanks for the information concerning our manuscript submitted to IJMS. The manuscript has been revised according to the comments from the reviewers, and the details are as follows:

Reviewer 1:

  1. I suggest authors to remove the phrase “Striking the” from the title.  

Reply: Thanks for the suggestion. In the modified version of manuscript, the phrase “Striking the” has been removed.

  1. What precursor did they use for PdO synthesis? 

Reply: Thanks for the comments. PdO used in this paper is commercially available. Also, we tried synthesis of PdO/g-C3N4 by pyrolysis Pd(OH)2/g-C3N4 as described in section 3.3.4 for 773 K at Argon atmosphere; however, the catalyst showed no activity for the selected reaction.

  1. Table 1, line 12 – why there are two different values for conversion, selectivity, and yield?

Reply: Thanks for the comments. The first value in each item stands for the catalyst was tested without 4Å molecular sieve, and the second value in each item stands for the catalyst was tested using 4Å molecular sieve to absorb water.

4 .    How was the selectivity to product resulted upon reduction of butyraldehyde? It seems Pd exhibited lower selectivity to -C=O reduced product. Did authors have any explanation in that? Chemoselectivity of Pt and Pd towards C=O and C=C reduction is widely reported in literature and authors should consider that. 

Reply: Thanks for the comments. Indeed, the -C=O can be reduced by Pd or Pt catalyst. However, in the previous trials we found that the selectivity of C=O reduction and reduction amination can be tuned by adjusting the ratio of amine to aldehyde and hydrogenation pressure. Decreasing the amine:aldehyde ratio and increasing the hydrogen pressure improve the selectivity of -C=O reduction. Thus, the ratio of amine to aldehyde 2:1 and a hydrogen pressure of 1.5 MPa were selected in this paper.

5 .    In Table 1, comparing line 12 and 13, it is apparent that when g-C3N4 support was used, activity of Pd(OH)2 was higher compared to Pd, however, the reverse is true when ACs was used as support. Can authors provide a rational in this regard? 

Reply: Thanks for the comments. According to literature investigation, several surface functional groups such as −O, −OH, and −COOH can be created on ACs by the oxidation treatments (nitric acid or water) which can stable Pd nano particles (ref.[38] in the revised version). In our opinion, we consider that the side reaction (i.e., aldol condensation) may proceed more easily on Pd/ACs catalyst whose support is treated by nitric acid or water since the aldol condensation can be catalyzed by acid sites. Consequently, the selectively of reductive amination for Pd/ACs is lower than that for Pd(OH)2/ACs but show higher reaction activity.

  1. Page 5, line 212 – I am not sure what authors meant for existence of H atom on Pd-oxide cluster? I think it should be the hydroxyl anion that are important for the activity. 

Reply: Thanks for the comments. According to the literatures, the fine structure of Pearlman’s catalyst which is written as Pd(OH)2/C consists of carbon supported (mostly) nano-particulate hydrous palladium oxide capped with a monolayer of hydroxyls hydrogen-bonded to a few layers of water: a core–shell structure of C/PdO/OH/H2O rather than uniform stoichiometric Pd(OH)2 (ref.[52] in the revised version). Thus, we think that the structure of home-made Pd(OH)2/g-C3N4 may possess similar structure and the H atom on Pd-oxide cluster matters much.

  1. Page 6, line 217 – I do not find any entry corresponding to 12f. authors should check that

Reply: Thanks for the comments. The “f” was labelled on the second value of Sel.% in the Entry 12.

  1. In Figure 1, authors mentioned that upon loading Pd, an increase in -OH stretching vibration was noted. In discussion, they should explicitly mention whether it is due to Pd or Pd(OH)2

Reply: Thanks for the comments. In the modified version of manuscript, the attribution of the O-H stretching has been emphasized, it belongs to Pd(OH)2.

  1. The presentation of Figure 1 is very confusing, there are both color code and a, b, c, d etc. which are very different. Authors should clarify how to follow Figure 1. 

Reply: Thanks for the comments. In the modified version of manuscript, “a, b, c, d” have been removed and the Pd loading per catalyst can be found in the legend.

  1. In figure 2c and 2d, what is the meaning of ACs-N and ACs-H? with AC as support, all the Pd(OH)2 catalyst show a peak corresponding to Pd(0) which suggested some reduction took place. Why is it so? Also, why the intensity is different across the catalysts?

Reply: Thanks for the comments. “ACs-N and ACs-H” stands for the ACs were treated by HNO3 and water, respectively, and it is further stated in the caption of Figure 2.

For the second question, in our opinion, the reduced palladium could be formed by the reducing influence of the carbonaceous support during catalyst preparation, which is in line with the results reported in the literature (ref.[52] in the revised version).

For third question, indeed, the intensity is different across the catalysts. We consider that the amounts of reducible groups such us −COOH may be different after different oxidation treatments (c-e):

©−COOH+ Pd2+→ © +Pd0+ H++ CO2

   In addition, using external reducing agent like hydrazine hydrate and the nature of the support (ACs vs g-C3N4) will change the peak intensity (a-b).

  1. The smaller particles showed in figure 4c are corresponding to Pd or Pd(OH)2 clusters? Did authors know what underlying reason for catalyst deactivation is, i.e., lowering both conversion and selectivity

Reply: Thanks for the comments. Based on the acquired results, we speculate that those smaller particles in figure 4c are likely to be Pd-nano clusters. The decrease of both the conversion and selectivity after several runs may be caused by leaching of the Pd nano-species.

  1. Why the authors used acetaldehyde instead of butyraldehyde for in-situ IR experiments?

Reply: Thanks for the comments. Based on previous trials, we think that the gasification temperature (>373K) of butyraldehyde is much higher than acetaldehyde that more severe aldol condensation may occur. Thus, acetaldehyde is used in the in-situ IR experiments since they have the same functional groups.

  1. Authors should move figure 7 to the supporting information

Reply: Thanks for the suggestion. Figure 7 has been moved to the supporting information.

  1. Line 425, it should be -OH instead of O-

Reply: Thanks for the comments. Since the hemiaminal formed in presence of O-Pd-OH site, according to theoretical calculations, we think that the expression “O-Pd-OH” may be more appropriate.

  1. In my opinion, hydrogen bonding is one of the major factor why Pd(OH)2 shows much higher activity compared to Pd. I suggest authors to consider the following paper and the back reference included in it. (Angew. Chem. Int. Ed. 10.1002/ange.202008178)

Reply: Thanks for the comments. The reference (Angew. Chem. Int. Ed. 10.1002/ange.202008178) has been added to this paper and a proper comment has been added.

  1. In Table 2, what are the major side products obtained when the selectivity to desired product is low

Reply: Thanks for the comments. The major side products obtained (Entry 11 and 12, Table 2) is aniline and benzene in presence of Pd-catalysts and hydrogen.

  1. Line 439-440, What authors meant hydroxyl group serves as a hydrogen transition state? Also, they should explain how presence of hydroxyl group facilitates dihydrogen activation. 

Reply: Thanks for the comments. We used “hydrogen transition site” rather than “hydrogen transition state” in line 439-440, and the hydroxyl group delivers hydrogen from H2 to the C=N group. To be more clear, here we changed “hydrogen transition site” to “hydrogen shuttle”.

As to the technical check comments sent by the editorial office, we have revised the manuscript format carefully according to the requirement.

In addition, all changes were marked with yellow color. We hope that the manuscript now is qualified for publication in IJMS.

Looking forward to your reply.

Thanks and best regards,

Shaodong Zhou

Reviewer 2 Report

In the manuscript “Striking the Underlying Mechanisms of Reductive Amination on Pd-catalysts: The Unique Role of Hydroxyl Group in Generating Sterically Hindered Amine”,  Zhou and colleagues report study on reductive amination of aldehydes with sterically demanding amines at Pd(OH)2/g-C3N4 catalyst. The authors compare the performance of Pd(OH)2/g-C3N4 with other Pd catalyst as well as with several non-Pd catalysts, discuss the mechanism of the reaction based on experimental and computational methods and report the results of reductive amination of aldehydes for a series of reactions.

The results might be interesting, but there are some issues to be addressed:

1) In conclusions, the authors state that “a highly selective Pd catalyst designed specifically for the preparation of sterically hindered amine is reported”. Yet, only chromatographic (GC) yields are reported. No isolation and purification procedure is given. What are the isolated yields of the hindered amines reported in Tables 1 and 2?

2) In eq. (3), the sum of molar amount of all products appears in the denominator. Please describe how were these products identified.

3) In section 2.5, the authors describe in situ FT-IR experiments as a series of desorption spectra of a mixture of diisopropylamine and acetaldehyde adsorbed at the catalyst surface by passing a gaseous mixture of the reactants through the sample wafer. This description is not consistent with the legends on Fig. 5, where separate spectra for the aldehyde are followed by the spectra of aldehyde/amine mixture followed by the spectra of aldehyde/amine/H2 mixture. I guess, the description in section 2.5 is incomplete. Please provide more detailed description of these experiments.

4) Are the reaction conditions of the in situ FT-IR experiments the same as the optimal conditions reported by the authors for their system? My impression from the description given in Section 2.5 is that the reaction conditions in the FT-IR experiments are significantly different from those reported in Table 2? Please comment on this.

5) What determined the choice of the time and the temperature reported in Tables 1 and 2? Please provide explanation in the text.

6) In the description of the reaction mechanism please use more accurate wordings. For example, the phrase “the hemiaminal intermediate is formed spontaneously without reaction barrier by simple proton exchange between diisopropylamine and a hydroxyl group (IM21 → IM22)” (p. 13, lines 421–423) is not an accurate description of the transformation of IM21 to IM22. There is no proton exchange between diisopropylamine and hydroxyl group during this transformation according to Scheme 1, and this is not the key structural change in this process. Please use more concise wording. Please check the rest of mechanistic considerations for the accuracy of descriptions.

7) Please provide more detailed explanation of the “unique physicochemical properties” (p. 2, line 74) of g-C3N4. They are crucial in the context of this manuscript. A good idea might be to compare the results of your work with the finding of Wang et al. that phenol can be selectively reduced to cyclohexanone at Pd@g-C3N4 (J. Am. Chem. Soc. 2011, 133, 2362–2365, DOI: 10.1021/ja109856y).

8) Please complement the description of the synthetic methods towards amines by reductive amination of carbonyl compounds with borohydride reagents. They have been implied in the synthesis of sterically hindered amines.

9) In Table 1, entries 10–16: are the numbers after decimal point relevant? Can it be just “1 wt%”?

10) Fig. 4: The inset on Fig.4c is unreadable. The legend attributes this inset to Fig. 4b. Please correct this.

11) P. 15, line 459: “of drops” – something is missing here.

12) Please check the manuscript for typos, there are some.

Author Response

Dear Editor,

Many thanks for the information concerning our manuscript submitted to IJMS. The manuscript has been revised according to the comments from the reviewers, and the details are as follows:

Reviewer 2:

  • In conclusions, the authors state that “a highly selective Pd catalyst designed specifically for the preparation of sterically hindered amine is reported”. Yet, only chromatographic (GC) yields are reported. No isolation and purification procedure is given. What are the isolated yields of the hindered amines reported in Tables 1 and 2?

Reply: Thanks for the comments. In the revised version of the manuscript, the isolated yields of the hindered amines reported in Tables 1 and 2 as well as the isolation procedure have been added.

  • In eq. (3), the sum of molar amount of all products appears in the denominator. Please describe how were these products identified.

Reply: Thanks for the comments. All products (i.e., sterically hindered amines, n-butanol and aldol condensation product) were identified by GC-MS.

  • In section 2.5, the authors describe in situ FT-IR experiments as a series of desorption spectra of a mixture of diisopropylamine and acetaldehyde adsorbed at the catalyst surface by passing a gaseous mixture of the reactants through the sample wafer. This description is not consistent with the legends on Fig. 5, where separate spectra for the aldehyde are followed by the spectra of aldehyde/amine mixture followed by the spectra of aldehyde/amine/H2 I guess, the description in section 2.5 is incomplete. Please provide more detailed description of these experiments.

Reply: Thanks for the comments. More detailed description of operation procedure for the in-situ FT-IR spectra has been provided in the revised version of manuscript.

  • Are the reaction conditions of the in situFT-IR experiments the same as the optimal conditions reported by the authors for their system? My impression from the description given in Section 2.5 is that the reaction conditions in the FT-IR experiments are significantly different from those reported in Table 2? Please comment on this.

Reply: Thanks for the comments. Indeed, the reaction conditions of the in-situ FT-IR experiments are not related with the optimal conditions (Table 1 or 2). This is because the substrates (i.e., amine and aldehyde) should be vaporized to the self-supported wafer to obtained absorption spectra. Determination of experimental conditions in Table 2 has been provided in the revised version of manuscript.

  • What determined the choice of the time and the temperature reported in Tables 1 and 2? Please provide explanation in the text.

Reply: Thanks for the comments. The explanation for choice of time, temperature and other reaction conditions has been provided in the revised version of manuscript.

  • In the description of the reaction mechanism please use more accurate wordings. For example, the phrase “the hemiaminal intermediate is formed spontaneously without reaction barrier by simple proton exchange between diisopropylamine and a hydroxyl group (IM21 → IM22)” (p. 13, lines 421–423) is not an accurate description of the transformation of IM21 to IM22. There is no proton exchange between diisopropylamine and hydroxyl group during this transformation according to Scheme 1, and this is not the key structural change in this process. Please use more concise wording. Please check the rest of mechanistic considerations for the accuracy of descriptions.

Reply: Thanks for the comments. The mechanism has been reconsidered and related statements have been improved in the revised version of the manuscript.

  • Please provide more detailed explanation of the “unique physicochemical properties” (p. 2, line 74) of g-C3N4. They are crucial in the context of this manuscript. A good idea might be to compare the results of your work with the finding of Wang et al. that phenol can be selectively reduced to cyclohexanone at Pd@g-C3N4(J. Am. Chem. Soc. 2011, 133, 2362–2365, DOI: 10.1021/ja109856y).

Reply: Thanks for the comments. The phrase “unique physicochemical properties” has been further explained and more comments on Wang et.al and our work has been provided in the revised version of the manuscript.

  • Please complement the description of the synthetic methods towards amines by reductive amination of carbonyl compounds with borohydride reagents. They have been implied in the synthesis of sterically hindered amines.

Reply: Thanks for the comments. The description of the synthetic methods towards amines by reductive amination of carbonyl compounds with borohydride reagents has complemented in the revised version of the manuscript.

  • In Table 1, entries 10–16: are the numbers after decimal point relevant? Can it be just “1 wt%”?

Reply: Thanks for the comments. No, the numbers after decimal point are not relevant with each other, and they reflect the degree of aggregation of palladium on different supports. In order to be more detailed for the authors, maybe it is better to keep them to reflect the exact palladium loading of per catalyst.

  • 4: The inset on Fig.4c is unreadable. The legend attributes this inset to Fig. 4b. Please correct this.

Reply: Thanks for the comments. The color of inset on Fig.4c has been changed to red and enlarged that can be more readable than its previous form.

  • P15, line 459: “of drops” – something is missing here.

Reply: Thanks for the suggestion. “of drops” has been corrected as “of target tertiary amine drops”.

  •  Please check the manuscript for typos, there are some.

Reply: Thanks for the suggestion. In the revised version of manuscript, some typos have been checked carefully and related expressions has been improved.

As to the technical check comments sent by the editorial office, we have revised the manuscript format carefully according to the requirement.

In addition, all changes were marked with yellow color. We hope that the manuscript now is qualified for publication in IJMS.

Looking forward to your reply.

Thanks and best regards,

Shaodong Zhou

Round 2

Reviewer 1 Report

Authors have provided satisfactory explanation to all the comments. I now recommend the manuscript for publication in the present journal.  

Author Response

Many thanks for your comments.

Reviewer 2 Report

The authors have partially addressed my concerns in the revised version. My comments to the reply of the authors:

“In conclusions, the authors state that “a highly selective Pd catalyst designed specifically for the preparation of sterically hindered amine is reported”. Yet, only chromatographic (GC) yields are reported. No isolation and purification procedure is given. What are the isolated yields of the hindered amines reported in Tables 1 and 2?

Reply: Thanks for the comments. In the revised version of the manuscript, the isolated yields of the hindered amines reported in Tables 1 and 2 as well as the isolation procedure have been added.”

Examination of Figs. S46, S45, S40 and S39 shows that at least butyldiisopropylamine and benzyldiisopropylamine are not quite purified, so the isolated yields in entry 3 (Table 2) and entry 12 (Table 1) are misleading.

A very interesting result is the exceptionally high isolated yield in several other entries. For example, the authors report that N,N-dibutyl-N-methylamine was isolated in 98% yield (entry 10, Table 2). This is a really impressive result, taking into account the reaction conditions: the authors take 2-fold molar excess of secondary amine with respect to the aldehyde and upon completion of the reaction isolate the product of reductive amination by vacuum distillation (Table 2, the legend). The boiling point of dibutylamine (which should still be present in the reaction mixture in ca. 1:1 ratio with the target amine) is 159 °C, the boiling point of N,N-dibutyl-N-methylamine is 165 °C. Please describe in details the experimental setup and the equipment which ensure such efficient separation of the compounds with so small difference in boiling points. This would be most useful for the practical application of your method.

“Please check the manuscript for typos, there are some.

Reply: Thanks for the suggestion. In the revised version of manuscript, some typos have been checked carefully and related expressions has been improved.”

Please check the manuscript for typos, there are some also in the revised fragments. In particular, p. 17, line 508 probably needs revision, as well as eq. 2 on the same page.

Author Response

Dear Editor,

Many thanks for the information concerning our manuscript submitted to IJMS. The manuscript has been revised according to the comments from the reviewers, and the details are as follows.

Reviewer 2#

Many thanks for your comments. As to the remaining problems you pointed out, we further checked the associated issues carefully, and the reply to the left, unanswered questions are:

Examination of Figs. S46, S45, S40 and S39 shows that at least butyldiisopropylamine and benzyldiisopropylamine are not quite purified, so the isolated yields in entry 3 (Table 2) and entry 12 (Table 1) are misleading.

Reply: Thanks for the comments. Indeed, the isolated butyldiisopropylamine (Figure S40 and S39) still contains some impurities and is difficult to be further purified; thus, after evaluating the amount of impurities the isolated yield in Entry 12 (Table 1) was revised. For benzyldiisopropylamine, we tried to purify it again, however, unsatisfactory purity was still obtained. Most likely, the unique properties of this product result in certain interactions with the impurities. Thus, we decided to check in more detail this entry in our future work and not to report it in the current manuscript; this entry has been deleted. In addition, in the Section 3.2, the information about NMR tests have been provided in the revised version of the manuscript.

A very interesting result is the exceptionally high isolated yield in several other entries. For example, the authors report that N,N-dibutyl-N-methylamine was isolated in 98% yield (entry 10, Table 2). This is a really impressive result, taking into account the reaction conditions: the authors take 2-fold molar excess of secondary amine with respect to the aldehyde and upon completion of the reaction isolate the product of reductive amination by vacuum distillation (Table 2, the legend). The boiling point of dibutylamine (which should still be present in the reaction mixture in ca. 1:1 ratio with the target amine) is 159 °C, the boiling point of N,N-dibutyl-N-methylamine is 165 °C. Please describe in details the experimental setup and the equipment which ensure such efficient separation of the compounds with so small difference in boiling points. This would be most useful for the practical application of your method.

Reply: Thank you very much for the comments. Indeed, we also think this protocol may be of high importance for the practical application. Here we used paraformaldehyde as the substrate with a molecular weight 90, and the actual molar ratio formaldehyde:amine is thus 1.5:1. Thus, upon completion of the reaction, the targe product is the only amine in the system and it can be easily separated via vacuum distillation in a rotary evaporator.

2、Please check the manuscript for typos, there are some also in the revised fragments. In particular, p. 17, line 508 probably needs revision, as well as eq. 2 on the same page.

Reply: Thanks for the comments. In the p. 17, line 508, has been revised as “Here, the interaction of acetaldehyde and diisopropylamine was taken as an example (Figure 5f), and the catalyst sample (25mg) was filled in prior into a self-supported wafer.”, and a few other typos have also been corrected.

In addition, all changes were marked with “Track Changes”. We hope that the manuscript now is qualified for publication in IJMS.

Looking forward to your reply.

Thanks and best regards,

Shaodong Zhou

Round 3

Reviewer 2 Report

The authors clarified that "the actual molar ratio formaldehyde:amine is thus 1.5:1. Thus, upon completion of the reaction, the targe product is the only amine in the system and it can be easily separated via vacuum distillation in a rotary evaporator". I could not find this information in the manuscript. Please mention this explicitly in the manuscript and provide the description of the experimental procedure in section 3.4 with the molar ratios of the reagents used.

Author Response

Dear Editor,

Many thanks for the information concerning our manuscript submitted to IJMS. The manuscript has been revised according to the comments from the reviewers, and the details are as follows:

Reviewer 2:

The authors clarified that "the actual molar ratio formaldehyde:amine is thus 1.5:1. Thus, upon completion of the reaction, the targe product is the only amine in the system and it can be easily separated via vacuum distillation in a rotary evaporator". I could not find this information in the manuscript. Please mention this explicitly in the manuscript and provide the description of the experimental procedure in section 3.4 with the molar ratios of the reagents used.

Reply: Thanks for the comments. In the current manuscript, the actual molar ratio of formaldehyde: amine has been described in the caption of Table 2. In addition, the description of the experimental procedure in each entry has been added in section 3.4.

The computational details and references in the SI have been moved into the manuscript.

In addition, all changes were marked with “Track changes”. We hope that the manuscript now is qualified for publication in IJMS.

Looking forward to your reply.

Thanks and best regards,

Shaodong Zhou